

# ABC transporters P-gp and Bcrp do not limit the brain uptake of the novel antipsychotic and anticonvulsant drug cannabidiol in mice

Natalia Brzozowska[1], Kong M. Li[1], Xiao Suo Wang[3], Jessica Booth[4], Jordyn Stuart[2,4], Iain S. McGregor[2,4] and Jonathon C. Arnold[1,2]

[1] Discipline of Pharmacology, School of Medical Science, University of Sydney, Sydney, NSW, Australia
[2] The Lambert Initiative of Cannabinoid Therapeutics, The Brain and Mind Centre, University of Sydney, Sydney, NSW, Australia
[3] Bosch Mass Spectrometry Facility, Bosch Institute, Sydney Medical School, University of Sydney, Sydney, NSW, Australia
[4] Psychopharmacology Laboratory, School of Psychology, Faculty of Science, University of Sydney, Sydney, NSW, Australia

Corresponding author
Jonathon C. Arnold,
jonathon.arnold@sydney.edu.au

## ABSTRACT

Cannabidiol (CBD) is currently being investigated as a novel therapeutic for the treatment of CNS disorders like schizophrenia and epilepsy. ABC transporters such as P-glycoprotein (P-gp) and breast cancer resistance protein (Bcrp) mediate pharmacoresistance in these disorders. P-gp and Bcrp are expressed at the blood brain barrier (BBB) and reduce the brain uptake of substrate drugs including various antipsychotics and anticonvulsants. It is therefore important to assess whether CBD is prone to treatment resistance mediated by P-gp and Bcrp. Moreover, it has become common practice in the drug development of CNS agents to screen against ABC transporters to help isolate lead compounds with optimal pharmacokinetic properties. The current study aimed to assess whether P-gp and Bcrp impacts the brain transport of CBD by comparing CBD tissue concentrations in wild-type (WT) mice versus mice devoid of ABC transporter genes. P-gp knockout ($Abcb1a/b^{-/-}$), Bcrp knockout ($Abcg2^{-/-}$), combined P-gp/Bcrp knockout ($Abcb1a/b^{-/-} Abcg2^{-/-}$) and WT mice were injected with CBD, before brain and plasma samples were collected at various time-points. CBD results were compared with the positive control risperidone and 9-hydroxy risperidone, antipsychotic drugs that are established ABC transporter substrates. Brain and plasma concentrations of CBD were not greater in P-gp, Bcrp or P-gp/Bcrp knockout mice than WT mice. In comparison, the brain/plasma concentration ratios of risperidone and 9-hydroxy risperidone were profoundly higher in P-gp knockout mice than WT mice. These results suggest that CBD is not a substrate of P-gp or Bcrp and may be free from the complication of reduced brain uptake by these transporters. Such findings provide favorable evidence for the therapeutic development of CBD in the treatment of various CNS disorders.

## INTRODUCTION

Cannabidiol (CBD), a non-psychoactive constituent of cannabis, displays much potential as a novel therapeutic treatment for various CNS disorders including schizophrenia and epilepsy (*Bumb, Enning & Leweke*, *2015*; *Iseger & Bossong*, *2015*; *Longo, Friedman & Devinsky*, *2015*). CBD has anticonvulsant and antipsychotic effects in animal models of epilepsy and schizophrenia (*Arnold, Boucher & Karl*, *2012*; *Jones et al.*, *2012*; *Jones et al.*, *2010*; *Mao et al.*, *2015*). Anecdotal reports and early clinical findings support CBD's ability to reduce seizure rates in humans with a good safety profile (*Rosenberg et al.*, *2015*). Phase 3 clinical trials are currently investigating CBD in the treatment of epilepsy, most notably in severe and pharmacoresistant childhood epilepsies. There is also promising evidence that CBD is a novel antipsychotic, with a phase 2 clinical trial showing CBD reduces symptoms in schizophrenia patients with comparable efficacy to a conventional antipsychotic drug without producing extrapyramidal side-effects, sedation or weight gain (*Bumb, Enning & Leweke*, *2015*; *Iseger & Bossong*, *2015*; *Leweke et al.*, *2012*). Numerous further clinical trials are currently underway examining the efficacy of CBD in treating schizophrenia. The precise pharmacodynamic mechanisms responsible for CBD's anticonvulsant and antipsychotic efficacy are hotly debated and may involve inhibition of degradation of the endocannabinoid anandamide. Although there are numerous other contenders as CBD is a promiscuous drug that interacts with multiple drug targets including G-protein-coupled receptor 55 (GPR55), transient receptor potential vanilloid type 1 (TRPV1) channels, and adenosine transporters (*Leweke et al.*, *2012*; *McPartland et al.*, *2015*; *Rosenberg et al.*, *2015*).

Resistance to treatment is a major stumbling block in the clinical management of epilepsy and schizophrenia. Approximately 30% of both schizophrenia and epilepsy patients do not respond adequately to drug therapy (*Hoosain et al.*, *2015*; *Van Os & Kapur*, *2009*) and ATP-binding cassette (ABC) transporters play a role in treatment-resistance (*Bebawy & Chetty*, *2008*; *Brandt et al.*, *2006*). ABC transporters are a large superfamily of proteins that actively transport substrates across biological membranes and thus influence the disposition of substrate drugs (*Hee Choi & Yu*, *2014*; *Kathawala et al.*, *2015*). The best characterized ABC transporters are P-glycoprotein (P-gp, *Abcb1*) and breast cancer resistance protein (Bcrp, *Abcg2*), which are both efflux pumps localized at various pharmacological barriers in the body including the blood brain barrier (*Löscher & Potschka*, *2005*). Many antipsychotic and anticonvulsant drugs are substrates of P-gp, which strongly limits the brain accumulation of these agents by extruding the drugs from the brain parenchyma back into the blood (*Boulton et al.*, *2002*; *Doran et al.*, *2005*; *Luna-Tortós, Fedrowitz & Löscher*, *2008*; *Zhang et al.*, *2010*). Few studies have examined the substrate profile of CNS drugs for Bcrp, although lamotrigine has recently been demonstrated to be a substrate of human and mouse Bcrp (*Römermann, Helmer & Löscher*, *2015*).

There is evidence that genetic variation in P-gp influences treatment response to antipsychotic and antiepileptic drugs such as olanzapine, risperidone, paliperidone (9-hydroxy risperidone, the active metabolite of risperidone), phenobarbital and phenytoin (*French*, *2013*; *Wolking et al.*, *2015*). Furthermore, P-gp and Bcrp are upregulated at the BBB in epilepsy and schizophrenia and contribute to pharmacoresistance by limiting

the brain uptake and efficacy of anticonvulsant and antipsychotic drugs (*Aronica et al.*, *2005*; *Bauer et al.*, *2014*; *De Klerk et al.*, *2010*; *Lazarowski et al.*, *2007*; *Van Vliet et al.*, *2005*). Drugs that are not substrates of ABC transporters will then make better therapeutics, as they will be immune to drug resistance mediated by these proteins. Indeed, it has become common practice in drug development to screen against ABC transporters to help isolate lead compounds that are less likely to fail in clinical trials due to suboptimal pharmacokinetic properties. It is therefore important to establish whether CBD is an ABC transporter substrate. We have shown that the main psychoactive constituent of cannabis, $\Delta^9$-tetrahydrocannabinol (THC) is a substrate of both P-gp and Bcrp (*Spiro et al.*, *2012*). CBD, an isomer of THC, inhibits both P-gp and Bcrp transport (*Feinshtein et al.*, *2013b*; *Holland et al.*, *2007*; *Zhu et al.*, *2006*), although results have not been consistent for P-gp (*Holland et al.*, *2006*). Inhibitors are often substrates so there is a need to clarify whether CBD is a substrate of P-gp or Bcrp and whether this has implications for the brain uptake of the compound.

The present study assesses whether CBD is a substrate of P-gp and Bcrp by utilising mice devoid of these ABC transporter genes singly or in combination. If CBD accumulates at greater concentrations in the brain of ABC transporter knockout animals than wild-type mice then this provides evidence that CBD is an ABC transporter substrate. The results of CBD will be compared with the positive controls risperidone and 9-hydroxy risperidone, as these antipsychotic drugs are established P-gp substrates (*Doran et al.*, *2005*). Our results will be useful in the evaluation of CBD as a therapeutic agent for CNS disorders from a pharmacokinetic perspective. If P-gp or Bcrp do not limit the brain concentrations of CBD, it implies that CBD might be free from drug resistance that is mediated by these ABC transporters.

## MATERIALS AND METHODS

### Animals

We used male wild-type (WT, FVB background strain), P-gp knockout ($Abcb1a/b^{-/-}$), Bcrp knockout ($Abcg2^{-/-}$) and combined P-gp/Bcrp knockout ($Abcb1a/b^{-/-} Abcg2^{-/-}$) mice aged between 4 and 5 months and weighing between 25 and 30 g (Taconic farms, New York, USA). P-gp, Bcrp and P-gp/Bcrp knockout mice were developed by Professor Alfred Schinkel and colleagues at the Netherlands Cancer Institute, Amsterdam (*Jonker et al.*, *2002*; *Schinkel et al.*, *1995*; *Schinkel et al.*, *1997*). Mice were housed in groups of 4–6 mice per cage and kept under a standard 12 h light/dark schedule. Food and water were available *ad libitum* and all cages contained various forms of environmental enrichment such as a mouse house igloo and running wheel, a paper roll, a climbing ring, tissue paper and sunflower seeds. The University of Sydney's Animal Ethics Committee approved all experimental procedures undertaken (Protocol number: K21/1-2013/3/5924) and all procedures were in accordance with the Australian Code of Practice for the Care and Use of Animals for Scientific Purposes.
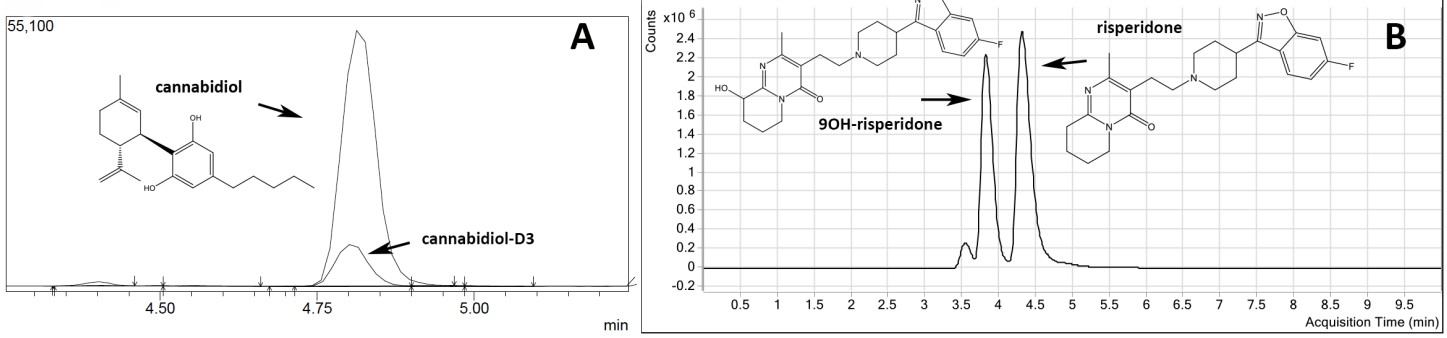

**Figure 1** **Representative chromatograms and molecular structures of tested compounds.** (A) CBD and internal standard CBD-D3 (B) Risperidone and 9-OH risperidone.

## Drug treatment

CBD (THC Pharm, Frankfurt, Germany) was dissolved in a mixture of ethanol, Tween 80, and saline (1:1:18) (*Long et al.*, *2013*; *Todd & Arnold*, *2016*) and administered via subcutaneous (s.c) injection at a dose of 10 mg/kg (*Doran et al.*, *2005*; *Pacchioni et al.*, *2010*). Risperidone (Sequoia Pharmaceuticals, Gaithersburg, MD, USA) was dissolved in a solution of 0.9% saline and 1% acetic acid and injected s.c. at 3 mg/kg. All drugs were freshly prepared before use and made at an injection volume of 10 ml/kg of body weight. At numerous time-points post-injection of CBD (1, 2 and 3 h) and risperidone (1 and 3 h), P-gp knockout, Bcrp knockout, P-gp/Bcrp knockout and WT mice were lightly anesthetised with isoflurane and blood collected via cardiac puncture. Blood samples were stored in ethylenediaminetetraacetic acid (EDTA) coated tubes to avoid coagulation and kept on ice before separation of plasma (*Spiro et al.*, *2012*). To separate the plasma from the blood, samples were centrifuged at 3,000 rpm for 10 min at 4 °C and the plasma collected in clean eppendorf tubes (*Wang et al.*, *2004*). The brains were immediately extracted and snap frozen in liquid nitrogen. Both the brain and plasma samples were stored at –80 °C before LC-MS/MS analysis.

## Quantification of CBD in brain and blood samples

CBD was extracted using a previously outlined method from our group (*Johnston et al.*, *2014*). In brief, a deuterated CBD-D$_3$ internal standard solution was added to every brain or plasma sample (see Fig. 1). Calibration standards and quality control (QC) samples were prepared by spiking drug-free mouse plasma or drug-free brain homogenates, at linear concentrations from 10 to 400 ng/g of CBD for brain analysis and 10–300 ng/ml of CBD for plasma analysis. The standards were vortexed and treated identically to other samples. Half brains were homogenised in dH$_2$0 at a 1:6 ratio (w/v) with 1 mL brain homogenate. For plasma analysis, 0.5 mL of a sample was used. Brain and plasma samples were prepared by slowly adding 2 mL ice-cold acetonitrile, mixed thoroughly and centrifuged at 3,000 rpm for 10 min. The acetonitrile was decanted into clean tubes and all samples were evaporated using the Genevac EZ-2 evaporation system for approximately 3–4 h. After reconstituting the samples with 2 mL dH$_2$0 the samples were loaded onto Styre Screen® SSTHC063

solid-phase extraction (SPE) columns (60 mg/3 ml) from United Chemical Technologies (Horsham, PA, USA). Columns were then washed with 1 mL water/acetonitrile/$NH_4OH$ (84:15:1) and dried thoroughly under vacuum (10 mm Hg) for 10–15 min. Samples were eluted from the column by adding 3 mL of hexane/ethyl acetate/glacial acetic acid (49:49:2). Extracts were completely dried under a nitrogen gas stream at 60 °C for 5–10 min and reconstituted with 50 µl initial mobile phase (40% methanol and 60% 10 mM ammonium acetate) for analysis. All quantification was performed using a Shimadzu 8030 triple quadrupole mass spectrometer. The mobile phase consisted of (A) 10 mM ammonium acetate in water and (B) methanol. The limits of quantification (LOQ) for plasma analysis were 1.5 ng/ml and 11.5 ng/g for brain analysis.

## Quantification of risperidone and 9-hydroxy risperidone in brain and blood samples

For plasma analysis, 10 µl of methyl-risperidone (10 µM) internal standard (IS) solution and 0.5 mL of $PO_4$ buffer (pH 5.0) were added to each 0.1 mL sample of plasma. Calibration standards were prepared by spiking drug-free mouse plasma at concentrations of 2–200 ng/ml for risperidone and 9-hydroxy risperidone. The standards were vortexed and treated identically to other plasma samples. For extraction, plasma samples underwent SPE using Varian SPEC 3 mL MP3 (15 mg) microcolumns from Agilent (Santa Clara, CA, USA). Columns were first conditioned by adding 0.5 mL methanol followed by 0.5 mL 0.1 M $PO_4$ buffer (pH 5.0). Plasma samples were then loaded onto the column. The columns were washed in 0.5 mL of acetic acid (1 M) and 0.5 mL methanol and then dried under vacuum for approximately 2 min. The samples were eluted from the column using 1 mL of freshly prepared dichloromethane/isopropanol/ammonia (80:20:2). The elutant was evaporated to dryness using a SpeedVac centrifugal evaporator. Samples were reconstituted with 200 µl of 50% acetonitrile.

For brain analysis, brains were dissected in half, weighed and one half of the brain was homogenised in $dH_2O$ at a 1:2 ratio ($w/v$). All samples and calibrators received 50 µl of 100 nM internal standard. Calibrators were made up of 1.5 mL of 0.1 M phosphate buffer (pH 6) spiked with linear concentrations of 1–100 nM of risperidone and 9-hydroxy risperidone. Sample homogenates were centrifuged at 14,000 g for 10 min and the supernatant collected. The brain supernatant underwent identical preparation and extraction as plasma samples and the elutant evaporated accordingly. Samples were reconstituted with 50 µl of mobile phase A. The linear gradient solutions consisted of mobile phase (A) 5 mM ammonium formate (pH 6) and (B) 90% acetonitrile. All quantification was performed using triple quadrupole liquid chromatography-mass spectrometry (Agilent 6460). The LOQ for brain analysis was 1.5 ng/g and 1 ng/ml for plasma analysis.

## Statistical analysis

Two-way ANOVA with factors of genotype (including WT, P-gp, Bcrp and P-gp/Bcrp knockout mice) and time were performed on brain and plasma concentrations as well as brain/plasma ratios of CBD, risperidone and 9-hydroxy risperidone. In the instance of finding an overall main effect of genotype or a genotype by time interaction, Tukey's post-hoc analysis was used to individually compare WT mice to P-gp knockout, Bcrp knockout,

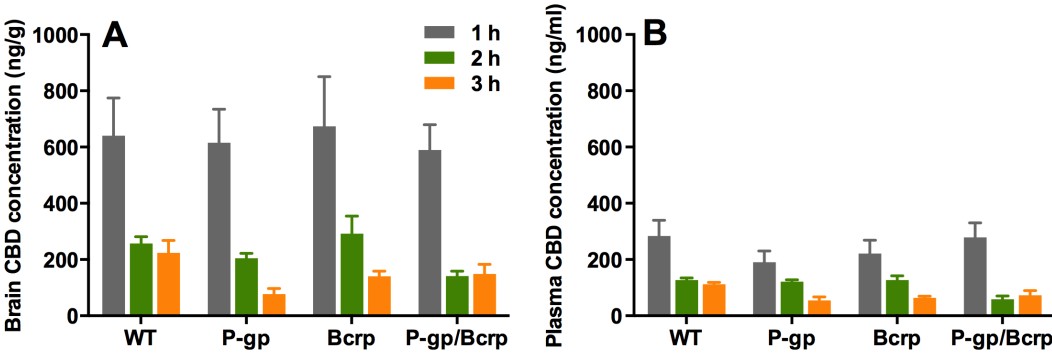

**Figure 2** **P-gp or Bcrp transporter knockout did not influence CBD brain and plasma concentrations.**
Brain and plasma CBD samples were collected from WT and ABC transporter knockout mice. Samples
were collected at 1, 2 and 3 h after an acute 10 mg/kg s.c. injection of CBD. (A) CBD brain concentration
(B) CBD plasma concentration. CBD, cannabidiol; WT, wild-type, P-gp, $Abcb1a/b^{(-/-)}$ (P-gp knockout);
Bcrp, $Abcg2^{(-/-)}$ (Bcrp knockout); P-gp/Bcrp, $Abcb1a/b^{(-/-)}Abcg^{(-/-)}$ (P-gp/Bcrp combined knockout).
Data represent mean + S.E.M.

or P-gp/Bcrp knockout mice. In addition, P-gp knockout mice were compared to P-gp/Bcrp
knockout mice. The latter comparison is important, as compensation with transporters
has been reported in knockout mice studies (*Tang et al.*, *2013*; *Vlaming et al.*, *2006*). For
example, double knockout of P-gp and Bcrp may significantly increase the brain uptake of
drugs when no altered disposition was observed in single P-gp knockout or Bcrp knockout
mice (*Tang et al.*, *2013*). This has been attributed to Bcrp being induced in response to P-gp
knockout and vice versa. Differences were deemed statistically significant when $P < 0.05$.

## RESULTS

### P-gp or Bcrp knockout did not increase brain or plasma concentrations of CBD

Representative chromatogram traces of CBD and the internal standard CBD-D3 are shown
in Fig. 1A. Concentrations of CBD in brain and plasma (Fig. 2) were not altered in P-gp,
Bcrp or P-gp/Bcrp knockout mice compared to WT mice as supported by overall two-way
ANOVA with no main effect of genotype or a genotype by time interaction. A significant
main effect of time was observed in both brain ($F(3, 60) = 30.5$, $P < 0.0001$) and plasma
($F(2, 63) = 34.8$, $P < 0.0001$) samples, with CBD concentrations decreasing over the three
hour time period in all genotypes. Tukey's post-hoc comparisons confirmed that no
significant differences were observed between WT when individually compared to P-gp
knockout, Bcrp knockout or P-gp/Bcrp knockout mice at 1, 2 or 3 h time-points. Nor were
there any significant differences between P-gp knockout and P-gp/Bcrp knockout mice at
any time-point.

### P-gp or Bcrp knockout did not influence CBD brain/plasma ratios, whereas P-gp knockout profoundly increased risperidone and 9-hydroxy risperidone brain/plasma ratios

No differences were observed between P-gp, Bcrp or P-gp/Bcrp knockout and WT mice (Fig.
3A) in their brain/plasma CBD concentration ratios (with overall a ratio of approximately
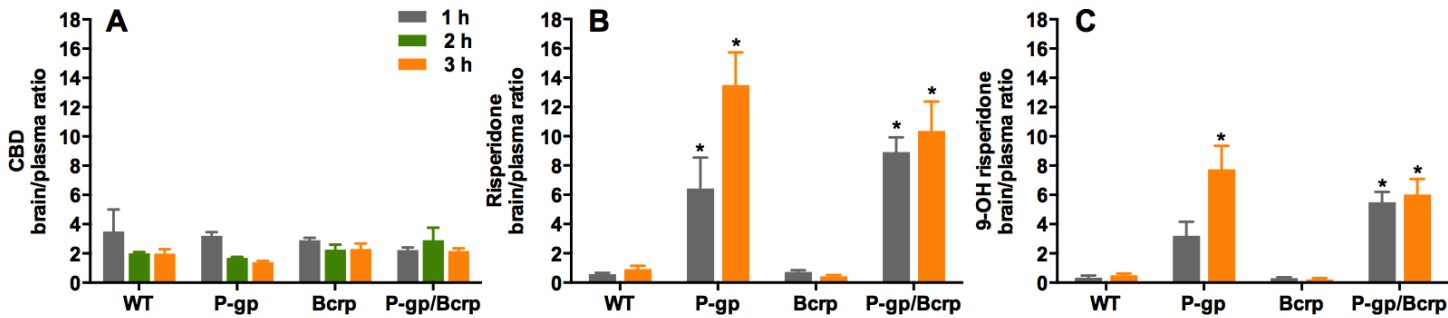

**Figure 3** **P-gp or Bcrp knockout did not alter CBD brain/plasma concentration ratios, whilst P-gp knockout profoundly increased risperidone and 9-hydroxy risperidone brain/plasma ratios.** (A) CBD brain/plasma concentration ratios (B) risperidone brain/plasma concentration ratios (C) 9-hydroxy risperidone brain/plasma concentration ratios. CBD, cannabidiol; 9-OH risperidone, 9-hydroxy risperidone; WT, wild-type; P-gp, $Abcb1a/b^{(-/-)}$ (P-gp knockout); Bcrp, $Abcg2^{(-/-)}$ (Bcrp knockout); P-gp/Bcrp, $Abcb1a/b^{(-/-)}Abcg^{(-/-)}$ (P-gp/Bcrp combined knockout). Tukey post-hoc analyses, *$p < 0.05$ for comparisons between WT and knockout mice. Data represent mean + S.E.M.

2), as two-way ANOVA showed no main effect of genotype and no genotype by time interaction. This was confirmed by post-hoc comparisons. As CBD did not show any altered disposition in ABC transporter knockout mice, we sought to show in our hands that other known transporter substrates, that is risperidone and 9-OH risperidone, display increased brain uptake in knockout animals. This would prove that our negative finding with CBD was not a false negative due to some procedural error.

Representative chromatogram traces of risperidone and 9-OH risperidone are shown in Fig. 1B. Two-way ANOVA indicated a significant overall main effect of genotype in the risperidone and 9-hydroxy risperidone brain/plasma concentration ratios ($F_{(3, 39)} = 34.62$, $P < 0.0001$; $F_{(3, 39)} = 32.17$, $P < 0.0001$ respectively) (Figs. 3B and 3C). The brain/plasma ratio for risperidone and 9-hydroxy risperidone also tended to increase over time in all genotypes (main effect of time: $F_{(1, 39)} = 5.66$, $P < 0.05$; $P_{(1, 39)} = 5.65$, $P < 0.05$ respectively). Tukey's post-hoc analysis showed P-gp knockout and P-gp/Bcrp knockout had significantly greater brain/plasma risperidone concentration ratios than WT mice at both 1 h and 3 h time points respectively ($Ps < 0.05$). In the P-gp knockout and P-gp/Bcrp knockout mice the brain/plasma risperidone ratios reached as high as 13 whereas the ratio for WT mice was less than 1. Brain/plasma concentration ratios of 9-hydroxy risperidone in P-gp knockout mice and P-gp/Bcrp knockout mice were significantly higher than the WT mice ratio at the 3 h time point ($Ps < 0.05$). In the P-gp knockout and P-gp/Bcrp knockout mice the brain/plasma 9-hydroxy risperidone ratio reached as high as 8 whereas the ratio for WT mice was less than 1. P-gp/Bcrp knockout mice achieved a significantly higher brain/plasma concentration ratio for 9-hydroxy risperidone than WT mice at 1 h, however P-gp knockout mice failed to reach significance when compared to WT mice at this timepoint. Although, P-gp knockout and P-gp/Bcrp knockout mice were not statistically different in their brain/plasma concentration ratios for risperidone and 9-hydroxy risperidone at either the 1 or 3 h timepoints implying no cooperation between P-gp and Bcrp in the transport of these drugs. Bcrp knockout mice did not display significantly different brain/plasma concentrations of risperidone and 9-hydroxy risperidone to WT at either timepoint.

## DISCUSSION

This study shows that the ABC transporters P-gp and Bcrp do not influence the brain uptake of CBD, a novel antipsychotic and anticonvulsant drug. P-gp, Bcrp and P-gp/Bcrp knockout mice did not accumulate greater brain or plasma concentrations of CBD compared to WT mice, nor were the brain/plasma concentration ratios of CBD influenced by knockout of the ABC transporter genes. By comparison, the known substrates of P-gp risperidone and 9-hydroxy risperidone (*Doran et al.*, *2005*; *Ejsing, Pedersen & Linnet*, *2005*; *Kirschbaum et al.*, *2008*; *Wang et al.*, *2004*), displayed increased brain/plasma concentration ratios in P-gp knockout mice compared to WT mice, reaching as high as approximately 13 and 8 times respectively, whereas WT mice only attained ratios <1. Taken together these results suggest P-gp strongly regulates the brain uptake of the antipsychotic drugs risperidone and 9-hydroxy risperidone but not CBD.

That P-gp and Bcrp did not influence the brain uptake of CBD is somewhat surprising given our prior research showing these transporters regulate the brain concentrations of THC, the main psychoactive constituent of cannabis (*Spiro et al.*, *2012*). CBD and THC are isomers with very similar lipophilicity. However, CBD is formed when the central pyran ring of THC is opened and the oxygen in the ring is converted into a free hydroxy group (*Compton et al.*, *1992*). This subtle chemical modification yields remarkable differences in the pharmacological activity of these drugs. For instance, THC is a partial agonist at $CB_1$ cannabinoid receptors and therefore elicits profound psychotropic effects, whilst CBD has poor affinity for the orthosteric site of this receptor and doesn't have appreciable psychoactivity (*Laprairie et al.*, *2015*; *Long et al.*, *2010*). CBD has a growing list of distinctive properties to THC. For example, CBD inhibits the anandamide degradative enzyme fatty acid amid hydrolase (FAAH) and activates TRPV1 receptors, actions not shared with THC (*McPartland et al.*, *2015*; *Pertwee*, *2008*). Our data here provides yet another example, this time of a differential substrate binding character for ABC transporter proteins.

Our finding that CBD is not a substrate of murine P-gp and Bcrp is consistent with *in vitro* data with cells expressing human transporters. CBD did not stimulate ATPase activity in insect membranes expressing human P-gp and Bcrp, unlike known P-gp and Bcrp substrates verapamil and sulphasalazine respectively (*Holland et al.*, *2007*; *Zhu et al.*, *2006*). Therefore it appears that our results here may generalize to human transporters, although future studies assessing CBD transport by human ABC transporters using transwell assays would strengthen this viewpoint. We and others have shown that both CBD and THC inhibit P-gp and Bcrp (*Feinshtein et al.*, *2013a*; *Feinshtein et al.*, *2013b*; *Holland et al.*, *2007*; *Zhu et al.*, *2006*). While THC appears to be a competitive substrate, as it is actively transported by P-gp and Bcrp (*Bonhomme-Faivre et al.*, *2008*; *Spiro et al.*, *2012*), CBD's ability to inhibit the transporters occurs in the absence of active transport. This phenomenon is not without precedent as paracetamol and haloperidol both inhibit P-gp but are not actively transported as substrates (*Feng et al.*, *2008*; *Novak et al.*, *2013*) and gefitinib inhibits Bcrp while not being a substrate (*Galetti et al.*, *2015*).

Future studies are needed to examine whether CBD's ability to inhibit ABC transporters alters the pharmacokinetics of co-administered drugs that are ABC transporter substrates

like risperidone. Interestingly co-administration of CBD with clobazam in children with refractory epilepsy increased plasma concentrations of clobazam and its active metabolite norclobazam by 60 and 500% respectively (*Geffrey et al.*, *2015*). Whether norclozabam is an ABC transporter substrate is unknown; however, clobazam is a duel P-gp/Bcrp substrate, opening the possibility that these transporters may contribute to this drug interaction (*Nakanishi et al.*, *2013*). In addition, CBD inhibition of ABC transporters appears relevant to pharmacokinetic interactions between CBD and THC, as THC is a dual P-gp and Bcrp substrate, and CBD potentiates some of the effects of THC via increasing brain THC concentrations (*Klein et al.*, *2011*; *Spiro et al.*, *2012*; *Todd & Arnold*, *2016*). Extended exposure to CBD has also been shown to influence the expression of P-gp in cancer cells that is mediated by CB2 and TRPV1 receptors (*Arnold et al.*, *2012*; *Holland et al.*, *2006*). Thus a future study assessing the impact of chronic CBD on P-gp expression in brain microvessels is warranted, as CBD-induced upregulation of P-gp may impact upon the pharmacokinetics of co-administered substrate drugs. Another limitation of the current study is that CBD metabolites were not examined, as no analytical standards for these compounds were commercially available. One metabolite of particular interest is 7-OH CBD which appears to be an active anticonvulsant with greater potency than CBD (*Jiang et al.*, *2011*; *Stott et al.*, *2015*; *Ujváry & Hanuš*, *2016*). Future studies need to address whether CBD metabolites like 7-OH CBD are ABC transporter substrates.

Our results suggest that risperidone and 9-hydroxy risperidone are not Bcrp substrates and that there is no cooperation between P-gp and Bcrp in the transport of these antipsychotic drugs. If these drugs were Bcrp substrates then the Bcrp knockout mice would have displayed greater brain and plasma concentrations of these agents than WT mice. Further, if these antipsychotic drugs were dual substrates of P-gp and Bcrp we would have expected the double P-gp/Bcrp knockout mice to display greater brain and plasma concentrations than P-gp knockout mice alone. Such observations have been made for other drugs, for example the Bcrp substrates prazosin and mitoxantrone display greater brain or plasma concentrations in Bcrp knockout mice than WT mice (*Cisternino et al.*, *2004*) and the dual P-gp and Bcrp substrates sunitinib and dasatinib show greater brain concentrations in P-gp/Bcrp knockout mice than P-gp or Bcrp knockout mice alone (*Tang et al.*, *2013*).

Cannabidiol is currently being assessed in randomized controlled trials as a novel antipsychotic and anticonvulsant agent, supported by an array of preclinical and human data (*Arnold, Boucher & Karl*, *2012*; *Devinsky et al.*, *2014*; *Leweke et al.*, *2012*). ABC transporters may play an important role in pharmacoresistance, which is a major stumbling block in the successful treatment of schizophrenia and epilepsy. Indeed the ABC transporter substrate binding character is routinely assessed in the development of novel CNS therapeutics to ensure adequate brain uptake of the drug e.g., blonanserin (*Inoue et al.*, *2012*). As can be seen here, the brain uptake of the commonly used antipsychotic drug risperidone and its active metabolite 9-hydroxy risperidone is profoundly limited by P-gp. Similarly many anticonvulsant drugs such as phenytoin, phenobarbital and clobazam are also ABC transporter substrates and subject to poor brain uptake (*Nakanishi et al.*, *2013*; *Zhang et al.*, *2010*). Moreover, single nucleotide polymorphisms (SNPs) in *MDR1* increase the risk of resistance or greater interindividual response to antiepileptic and

antipsychotic drugs (*Bozina et al.*, *2008*; *Li et al.*, *2014*; *Shaheen et al.*, *2014*; *Vijayan et al.*, *2012*). Our data support that CBD may be free from the complication of reduced brain uptake or varied interindividual response to drug therapy, at least that is mediated by the ABC transporters P-gp and Bcrp. These findings provide evidence for the favourable pharmacokinetic properties of CBD in the treatment of CNS disorders and help build the case for the development of CBD as a therapeutic agent.

### Abbreviations

| | |
|---|---|
| **CBD** | Cannabidiol |
| **THC** | $\Delta^9$-tetrahydrocannabinol |
| **ABC** | Adenosine triphosphate (ATP) binding cassette |
| **P-gp** | P-glycoprotein |
| **Bcrp** | Breast cancer resistance protein |
| **WT** | Wild-type |
| **9-OH risperidone** | 9-hydroxy risperidone |
| **BBB** | Blood brain barrier |
| **FVB** | Friend virus B-type mice |
| **s.c.** | Subcutaneous |
| **EDTA** | Ethylenediaminetetraacetic acid |
| **LC-MS/MS** | Liquid chromatography and tandem mass spectrometry (triple quadrupole mass spectrometer) |
| **QC** | Quality control |
| **SPE** | Solid phase extraction |
| **LOQ** | Limits of quantification |
| **CB$_1$** | Cannabinoid 1 receptor |

### Funding

This work was supported by a University of Sydney Bridging Grant and a Bosch Translational Grant-in-Aid to JCA. JCA was also supported by the Brain & Behaviour Research Foundation (formerly National Alliance for Research on Schizophrenia and Depression: Young Investigator Award). NB is supported by an Australian Postgraduate Award Scholarship. The funders had no role in study design, data collection and analysis, decision to publish, or preparation of the manuscript.

### Grant Disclosures

The following grant information was disclosed by the authors:
University of Sydney Bridging Grant.
Bosch Translational Grant-in-Aid.
Brain & Behaviour Research Foundation.
Australian Postgraduate Award Scholarship.

### Competing Interests

The authors declare there are no competing interests.

## Author Contributions

- Natalia Brzozowska conceived and designed the experiments, performed the experiments, analyzed the data, wrote the paper, prepared figures and/or tables.
- Kong M. Li, Xiao Suo Wang, Jessica Booth, Jordyn Stuart and Iain S. McGregor contributed reagents/materials/analysis tools.
- Jonathon C. Arnold conceived and designed the experiments, wrote the paper, prepared figures and/or tables, reviewed drafts of the paper.

## Animal Ethics

The following information was supplied relating to ethical approvals (i.e., approving body and any reference numbers):

All experiments were approved by the University of Sydney Animal Ethics Committee in accordance with the Australian Code of Practice for the Care and Use of Animals for Scientific Purposes and ARRIVE guidelines.

Protocol Number: K21/1-2013/3/5924

Guideline Number GL003.

## Data Availability

The raw data has been supplied as Data S1 and S2.

## Supplemental Information

Supplemental information for this article can be found online at http://dx.doi.org/10.7717/peerj.2081#supplemental-information.

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
