# Peer review of "ABC transporters P-gp and Bcrp do not limit the brain uptake of the novel antipsychotic and anticonvulsant drug cannabidiol in mice"

_PeerJ, doi:10.7717/peerj.2081_

## Round 0.1 · original submission · Major Revisions

This work used P-gp knockout Abcb1a/b (-/-), Bcrp knockout Abcg2 (-/-), combined P-gp/Bcrp knockout Abcb1a/b(-/-)Abcg2(-/-) and WT mice to examine brain transport of cannabidiol (CBD) and used risperidone and 9-hydroxy risperidone as “positive controls” by measuring their brain and plasma concentrations. In general, the work is well-designed and performed, and the use of knockout animals and LC-MS/MS analysis of drugs are plus. All three reviewers recognize the positive parts of the study but also raised a number of questions which need to be addressed. My additional comments are as follows:

1. Check statistics and use statistic symbols correctly:

Figure 2: “*p<0.05, **p<0.01, ***p<0.001 for comparisons between WT and knockout mice”. However, the way the data presented can be easily confused as differences between 1 h and 3 h. It should be the differences between WT and knockout mice at 1 h time points, and at 3 h time point. In this case the statistic symbol should be put on the individual column, rather covering two columns. To make the figures simple and to the point, *p<0.05 would be sufficient.

Figure 2: “##p<0.01 for comparison between P-gp knockout and P-gp/Bcrp combined knockout”. However, (1) from the Figure B with the label, such a difference is not evident, from Figure C, there are differences but without labels. (2) What are the purpose and significance for such a comparison?
.
Figure 3: “*p<0.05, **p<0.01, ***p<0.001 for comparisons between WT and knockout mice”. Again, the symbol should be labeled on individual columns, as compares to WT mice, not covering two columns easily confused as comparisons between 1 h and 3 h.
2. Since the drug quant illation is the main stone of this study. In reference method (Johnston et al., 2014), there is no representative chromatography presented. The representative pictures of cannabidiol (CBD), risperidone and 9-hydroxy risperidone are desired, as least as supplementary files.

3. Reference citation and list should follow PeerJ format (e.g, journal names, two and three author rules) throughout the manuscript.

·

Basic reporting

The authors present interesting data about distinct properties of CBD compared with other common antipsychotics and anticonvulsants. The authors report that ABC transporters P-gp and Bcrp do not influence the brain uptake of CBD, a novel antipsychotic and anticonvulsant drug which provide evidence for the favourable pharmacokinetic properties of CBD in the treatment of CNS disorders and help build the case for the development of CBD as a therapeutic agent. The manuscript is well written, has important messages, and should be of great interest to the readers. However, there are still some issues need to be revised. In general, I have some comments and suggestions to the authors listed below.

Experimental design

Introduction:
1. The introduction provides a good, generalized background of the topic that quickly gives the reader a clear purpose of this study. However, to make the introduction
more substantial, the author may wish to provide several references to substantiate the possible underlying mechanisms of anticonvulsant and antipsychotic effects.
2. Please describe more about ABC transporter protein family.
3. Add more description about properties of P-gp and Bcrp (eg. molecular structure).
4. “There is evidence that genetic variation in P-gp influences treatment response to
antipsychotic and antiepileptic drugs…” Are these genetic variations related to specific diseases? Please specify.
5. Are Bcrp will be upregulated similar to P-gp during epilepsy and schizophrenia?
6. “If CBD were to accumulate at greater levels in the brain of knockout animals, then this provides evidence that it is an ABC transporter substrate.” It is not accurate. It should be compared with wild-type animals under the same experimental conditions. There is possibility that the transport of CBD has nothing to do with ABC transporter.
7. It is necessary to add the summary, purpose and influence of your current research at the end of Introduction part.

Materials and methods:
Generally, the experimental apparatus is quite standard, and is appropriate for the study.
1. In Animal part, please add light condition in order to maintain day/light cycle.
2. Please add references substantiate a. s.c. injection for all of the drugs (why not i.v.?); b. blood collection method; c. separation of plasma from the blood.
3. What is the post-hoc for two-way ANOVA?

Validity of the findings

Results:
Basically, the description of Results section is too simple, more detailed information should be added into Results section. It is not good that Results section look likes figure legends.
1. Please add the chemical structures of THC, CBD, risperidone and 9-hydroxy risperidone.
2. In Figure 1 legend, is it means + SEM or means ± SEM? The same as Figures 2 and 3.
3. In Figure 2, panel A, it does not look like that Bcrp has significant difference from the wild-type from the bar graph.
4. In Figure 3, both panels B and C do not have brain/plasma ratios from 2h time point. Please be consistent with panel A.

Discussion:
1. “CBD has a growing list of distinctive properties to THC”, please list more.
2. Please list current method in order to avoid resistance from ABC transporters or other related transporters if applicable.
3. There are no limitations of the study discussed. Think about what are the major limitations that should be discussed in the manuscript.
4. Need to write future directions of current research.

Additional comments

My recommendation:
Due to the suggestions listed above, I would suggest that this manuscript need to be revised with minor revision before getting accepted.

Reviewer 2 ·

Basic reporting

The paper applied LC-MS/MS analysis to measure CBD concentrations in blood and plasma in Pgp and BCRP knockout mice. The author concluded that CBD is not a substrate of Pgp and BCRP. CBD, as a drug for schizophrenia and epilepsy, is already being tested in clinical trials. The drug resistance has not been reported as a serious issue for CBD although it could be a problem for other epileptic drug candidates. Therefore, the significance of this study is less important. Also, the paper used only one assay- LC-MS to complete the study, which was not mechanistic, but rather descriptive, so it carries less weight.

Experimental design

There are a few questions regarding the figures.

1) In Figure 1 and 3, three time points were reported, but in Figure 2, only 2 time pointes were reported. Why?
2) In Figure 2, both compounds tested have been already established as ABC transporter substrates, so the findings were not novel. Since authors used them as positive controls, they should not be reported in a separate figure, but should be compared to CBD side by side.
3) In Figure, 3, the ratio of brain and plasma could be useful in some way, but all the values were already reported in previous figures. So, this figure is not new date and authors should do either way.

Validity of the findings

In general, I do not recommend publishing as Research Article since this paper presented negative results and data were not organized well. However, the journal may consider Short Communication if needed.

Additional comments

The paper applied LC-MS/MS analysis to measure CBD concentrations in blood and plasma in Pgp and BCRP knockout mice. The author concluded that CBD is not a substrate of Pgp and BCRP. CBD, as a drug for schizophrenia and epilepsy, is already being tested in clinical trials. The drug resistance has not been reported as a serious issue for CBD although it could be a problem for other epileptic drug candidates. Therefore, the significance of this study is less important. Also, the paper used only one assay- LC-MS to complete the study, which was not mechanistic, but rather descriptive, so it carries less weight. There are a few questions regarding the figures.

1) In Figure 1 and 3, three time points were reported, but in Figure 2, only 2 time pointes were reported. Why?
2) In Figure 2, both compounds tested have been already established as ABC transporter substrates, so the findings were not novel. Since authors used them as positive controls, they should not be reported in a separate figure, but should be compared to CBD side by side.
3) In Figure, 3, the ratio of brain and plasma could be useful in some way, but all the values were already reported in previous figures. So, this figure is not new date and authors should do either way.

Reviewer 3 ·

Basic reporting

This is a clear and well written manuscript. The background info and literature citation are appropriate.

Experimental design

The experimental design is straightforward and the data is clear cut. Meanwhile the results from double knockout mice strengthen the finding.

Validity of the findings

This study well demonstrate the CBD is not the substrate for the P-gp and Bcrp transporters in vivo. The finding will help the usage of CBD in therapy in the future.

Additional comments

Brzozowska and colleagues used P-gp, Bcrp and double knockout mice to demonstrate that CBD is not the substrate of these ABC transporters. This paper is well-written, the idea is clear and the discussion covers most of the concerns of the study. I have only minor concerns for consideration.
1. In the introduction line 106: It is therefore important to establish whether CBD is an ABC transporter substrate. Here the author proved that CBD is not the substrate for P-gp and Bcrp, but not sure how about other ABC transporter, CBD may have effects at different time point.
2. In the discussion section, the authors and other groups have found that the CBD could inhibit P-gp and Bcrp (line 268-270), although CBD is not the substrate of these transporters, but it could change the expression of these transporters. So, the authors may consider the long-term study for CBD treatment, and combine with resperidone, the result may similar to P-gp KO. Therefore, is CBD safe for long-term use in patients?
3. Spell out the THC in abbreviation and introduction sections.

---

## Round 0.2 · accepted · Accept

Thank you for your contribution to PeerJ. We are look forward to future contributions from you.

·

Basic reporting

The authors did a good job based on the comments for the revision of this manuscript.

Experimental design

The authors did a good job based on the comments for the revision of this manuscript.

Validity of the findings

The authors did a good job based on the comments for the revision of this manuscript.

Additional comments

The authors did a good job based on the comments for the revision of this manuscript.

Reviewer 2 ·

Basic reporting

The authors have address most of reviewers' comments.

Experimental design

The authors have address most of reviewers' comments.

Validity of the findings

The authors have address most of reviewers' comments.

Additional comments

The authors have address most of reviewers' comments.

Reviewer 3 ·

Basic reporting

This manuscript is well written and the result clear demonstrate their hypothesis

Experimental design

The experimental design is straightforward, and the data is clear.

Validity of the findings

The finding could also provide the evidence for the therapy development using CBD in the future.

Additional comments

This study is clearly demonstrate the idea that CBD is not the substrate of ABC transporters, P-gp and Bcrp. The models they used support their hypothesis and the results could provide the in vivo evidence for future therapy development using CBD.

---

## Author Rebuttal · Round 0.2

**Jonathon Arnold**
Associate Professor
Discipline of Pharmacology &
The Brain and Mind Research Institute

10 April 2016

Jie Liu
Academic Editor for *PeerJ*

Dear Jie Liu,

Thank you for reviewing our manuscript entitled "**ABC transporters P-gp and Bcrp do not limit the brain uptake of cannabidiol in mice**" by NI Brzozowska, KM Li, XS Wang, J Booth, J Stuart, IS McGregor and JC Arnold. We have responded to each comment one by one and modified the manuscript accordingly (see below). We hope you now find our manuscript suitable for publication.

Yours faithfully,

Jonathon Arnold

Discipline of Pharmacology &
Brain and Mind Research Institute
Sydney Medical School
Room 307, Blackburn building D06
The University of Sydney
NSW 2006 Australia

**T** +61 2 9351 6954
**F** +61 2 9351 2658
**E** jonathon.arnold@sydney.edu.au
**sydney.edu.au**

ABN 15 211 513 464
CRICOS 00026A

## Response to Editor's comments:

**1) Check statistics and use statistic symbols correctly:**

**Figure 2: "*p<0.05, **p<0.01, ***p<0.001 for comparisons between WT and knockout mice". However, the way the data presented can be easily confused as differences between 1 h and 3 h. It should be the differences between WT and knockout mice at 1 h time points, and at 3 h time point. In this case the statistic symbol should be put on the individual column, rather covering two columns. To make the figures simple and to the point, *p<0.05 would be sufficient.**

**RESPONSE:** In light of the comments of Reviewer 2 (2), we have removed Figure 2. We have applied this logic now to Figure 3 as requested [see *Results* (page 13)].

**Figure 2: "##p<0.01 for comparison between P-gp knockout and P-gp/Bcrp combined knockout". However, (1) from the Figure B with the label, such a difference is not evident, from Figure C, there are differences but without labels. (2) What are the purpose and significance for such a comparison?**

**RESPONSE:** In light of the comments of Reviewer 2 we have removed Figure 2. The purpose of this comparison as it now pertains to Figure 3 will be more clearly described in the manuscript. This comparison is important as there is often compensation with transporters or (Tang et al., 2013; Vlaming et al., 2006). That is, risperidone from our results does not appear to be a substrate of Bcrp, however the Bcrp knockout may induce P-gp which nullifies any effect of the Bcrp knockout. The only way to know whether this occurs is to assess in the double P-gp/Bcrp knockout. If there were compensation we would anticipate that the brain concentration of risperidone to be significantly greater in the P-gp/Bcrp knockout compared to the P-gp knockout. We have now made a clear justification for this comparison in the Statistics section [see *Statistical Analysis* (page 10), lines 3-12].

**Figure 3: "*p<0.05, **p<0.01, ***p<0.001 for comparisons between WT and knockout mice". Again, the symbol should be labeled on individual columns, as compares to WT mice, not covering two columns easily confused as comparisons between 1 h and 3 h.**

**RESPONSE:** We have retained Figure 3 and presented the data now in line with the Editor's suggestion. That is, individual comparisons between WT and knockout mice are now made at each specific timepoint [see *Results* (page 13)].

[Figure]

**2) Since the drug quant illation is the main stone of this study. In reference method (Johnston et al., 2014), there is no representative chromatography presented. The representative pictures of cannabidiol (CBD), risperidone and 9-hydroxy risperidone are desired, as least as supplementary files.**

RESPONSE: We have now added these traces as requested [see Figure 1. *Results* (page 10)].

**3) Reference citation and list should follow PeerJ format (e.g, journal names, two and three author rules) throughout the manuscript.**

RESPONSE: Citations and references have been amended to follow PeerJ format.

## Responses to Reviewer 1 comments:

**Experimental design**

**1) The introduction provides a good, generalized background of the topic that quickly gives the reader a clear purpose of this study. However, to make the introduction more substantial, the author may wish to provide several references to substantiate the possible underlying mechanisms of anticonvulsant and antipsychotic effects.**

RESPONSE: We have added this extra detail as requested [see *Introduction* (page 4), lines 14-20].

**2) Please describe more about ABC transporter protein family.**

RESPONSE: We have attempted to add more on the ABC transporter family in the Introduction [see *Introduction* (page 5), lines 1-12].

**3) Add more description about properties of P-gp and Bcrp (eg. molecular structure).**

RESPONSE: We respectfully disagree that adding anything on the molecular structure of these transporters is appropriate to this paper as it is not biophysical in nature.

**4) "There is evidence that genetic variation in P-gp influences treatment response to antipsychotic and antiepileptic drugs…" Are these genetic variations related to specific diseases? Please specify.**

**RESPONSE:** The variation isn't necessarily disease specific, some people just happen to have SNPs in these genes that makes them respond differently to drugs that are influenced by the gene's function. The fact the sentence refers to antipsychotic and anticonvulsant drugs implies that this is within the context of schizophrenia and epilepsy respectively. We do not agree it is necessary to specify this.

**5) Are Bcrp will be upregulated similar to P-gp during epilepsy and schizophrenia?**

**RESPONSE:** To the best of our knowledge there is no current evidence that Bcrp is upregulated in schizophrenia. However there is some evidence that Bcrp is upregulated in refractory epilepsy. We now refer to this research in the manuscript [see *Introduction* (page 5), lines 16-19].

**6) "If CBD were to accumulate at greater levels in the brain of knockout animals, then this provides evidence that it is an ABC transporter substrate." It is not accurate. It should be compared with wild-type animals under the same experimental conditions. There is possibility that the transport of CBD has nothing to do with ABC transporter.**

**RESPONSE:** We have modified the sentence to be stated more accurately [see *Introduction* (page 6), lines 7-10].

**7) It is necessary to add the summary, purpose and influence of your current research at the end of Introduction part.**

**RESPONSE:** We have modified the manuscript accordingly [see *Introduction* (page 6), lines 6-15].

**Materials and methods**

**1) In Animal part, please add light condition in order to maintain day/light cycle.**

**RESPONSE:** We have corrected the text to include the light condition [see *Animals* (page 6), line 22-23].

**2) Please add references to substantiate a. s.c. injection for all of the drugs (why not i.v.?); b. blood collection method; c. separation of plasma from the blood.**

**Response:** We have included references to justify these methods [see *Drug treatment* (page 7), lines 8, 16 and 18]. S.C injection was chosen over i.p. simply

because this mode of administration is much simpler and less prone to error than i.p. injection in mice.

**3) What is the post-hoc for two-way ANOVA?**

**Response:** We have now applied Tukey's post-hoc analysis as requested [see *Statistical Analysis* (page 10)].

**Validity of the findings**

**1) Please add the chemical structures of THC, CBD, risperidone and 9-hydroxy risperidone.**

**RESPONSE:** We have added the CBD, risperidone and 9-hydroxy risperidone chemical structures as requested in Figure 1 [see Figure 1 *Results* (page 10)].

**2) In Figure 1 legend, is it means + SEM or means ± SEM? The same as Figures 2 and 3.**

**RESPONSE:** It is + SEM and this was described in Figure's 1, 2 and 3 in the submitted paper.

**3) In Figure 2, panel A, it does not look like that Bcrp has significant difference from the wild-type from the bar graph.**

**Response:** We have removed Figure 2 from the paper in light of comments by Reviewer 2 (2). When the data are expressed in brain/plasma concentrations the subtle decrease in brain risperidone concentrations in the Bcrp knockout mice disappears, so the finding was not robust.

**4) In Figure 3, both panels B and C do not have brain/plasma ratios from 2h time point. Please be consistent with panel A**.

**RESPONSE:** The point of the risperidone and 9-OH risperidone data is to provide a positive control to compare to our CBD results. As we could not show any differences in brain or plasma CBD concentrations between WT and knockout mice we wanted to prove this wasn't a false negative. By showing in our own hands and under the same conditions that risperidone and its metabolite are significantly increased in P-gp knockout mice we have proven our CBD results are indeed negative results and not explained by some procedural error. The results with risperidone and 9-OH risperidone have been demonstrated repeatedly in P-gp knockout mice (Doran et al., 2005; Ejsing, Pedersen & Linnet 2005; Kirschbaum et al., 2008; Wang et al., 2004) so we saw no reason to add a 2 h time-point for the sake of symmetry. By doing this we have achieved a reduction from an animal ethics perspective, using the least number of animals to demonstrate our scientific point.

[Figure]

**Discussion**

**1) "CBD has a growing list of distinctive properties to THC", please list more.**

**RESPONSE:** We have listed a couple more examples as suggested [see *Discussion* (page 14), lines 5-7].

**2) Please list current method in order to avoid resistance from ABC transporters or other related transporters if applicable.**

**RESPONSE:** There are no current methods, apart from using drugs that aren't susceptible to ABC transport.

**3) There are no limitations of the study discussed. Think about what are the major limitations that should be discussed in the manuscript.**

**RESPONSE:** In the discussion we have considered limitations and future directions of this research [see *Discussion* (page 14), lines 12-15 and 23-25, (page 15), lines 1-18]. Namely, we have stated that such research has been carried out in mice and to strengthen these findings, there is a need to assess CBD transport by human ABC transporters using transwell assays. We have also included an additional paragraph on limitations and future directions which incorporates comments also made by Reviewer 3 (2).

**4) Need to write future directions of current research.**

**Response:** See previous response, limitations have been followed up with future directions of this research [see *Discussion* (page 14) lines 12-15 and 23-25, (page 15) lines 1-18].

## Responses to Reviewer 2 comments:

**1) In Figure 1 and 3, three time points were reported, but in Figure 2, only 2 time pointes were reported. Why?**

**RESPONSE:** see response to Reviewer 1 – Validity of findings (4).

**2) In Figure 2, both compounds tested have been already established as ABC transporter substrates, so the findings were not novel. Since authors used them as positive controls, they should not be reported in a separate figure, but should be compared to CBD side by side.**

**RESPONSE:** We have decided to remove Figure 2 in line with the reviewer's comments. Reporting these data in addition to the Figure 3 data provides no additionally important information, especially given that the risperidone and metabolite findings have been published on numerous occasions elsewhere (Doran et al. 2005; Ejsing, Pedersen & Linnet 2005; Kirschbaum et al. 2008; Wang et al. 2004)**.** We have retained Figure 3 as it compares CBD to the positive control data side by side as requested.

We agree that the risperidone findings aren't completely novel, however there is new data in that no prior study has examined whether risperidone and 9-OH risperidone are regulated by Bcrp or the P-gp/Bcrp combination in knockout mice. This provides novel information, albeit negative findings that risperidone and 9-OH risperidone are unlikely Bcrp substrates and that P-gp and Bcrp do not cooperate to regulate the brain concentrations of these drugs.

**3) In Figure, 3, the ratio of brain and plasma could be useful in some way, but all the values were already reported in previous figures. So, this figure is not new date and authors should do either way.**

**RESPONSE:** See above, we have removed Figure 2. We disagree though that Figure 1 should be removed. Few papers have analysed CBD brain and plasma concentrations following CBD administration, so providing the actual concentrations is important new information. Moreover the time-course aspect of the data for CBD is lost if Figure 1 were to be removed (ie. the CBD concentration diminishes considerably over the 3 h sampling period).

**Validity of the findings**
**In general, I do not recommend publishing as Research Article since this paper presented negative results and data were not organized well. However, the journal may consider Short Communication if needed.**

**RESPONSE:** We strongly disagree that the paper be rejected based solely on the presentation of negative findings. This contributes to the file drawer problem in science. Some negative findings have important implications, such as is the case here, where CBD is not a P-gp or Bcrp substrate. Many drugs brain uptake is impaired by these transporters or complicated by interindividual variation in transport. This does not appear to be the case for CBD and this is favourable for its drug development for CNS disorders.

We have attempted to better organise the data in line with the reviewer's suggestions above.

## Responses to reviewer 3 comments:

**1) In the introduction line 106: It is therefore important to establish whether CBD is an ABC transporter substrate. Here the author proved that CBD is not the substrate for P-gp and Bcrp, but not sure how about other ABC transporter, CBD may have effects at different time point.**

**RESPONSE:** Our findings can only be limited to P-gp and Bcrp. While compensation has been reported for some drugs, analysis of the transporter proteome in ABC transporter mice shows no evidence of altered expression of other transporters in brain microvessels (Agarwal et al., 2012).

As can be seen in Figure 1 the CBD concentrations in the brain and blood are diminishing rapidly and are likely to be much lower if not undetectable at later time-points. We therefore think it unlikely that any change would be evident at these later timepoints and even if there were a change is would be subtle and of little significance when considering the whole area under the concentration-time curve.

**2) In the discussion section, the authors and other groups have found that the CBD could inhibit P-gp and Bcrp (line 268-270), although CBD is not the substrate of these transporters, but it could change the expression of these transporters. So, the authors may consider the long-term study for CBD treatment, and combine with resperidone, the result may similar to P-gp KO. Therefore, is CBD safe for long-term use in patients?**

**RESPONSE:** We think this is an excellent suggestion and have included it as a future study in the Discussion section [see *Discussion* (page 14), lines 23-25, (page 15), lines 1-13].

**3) Spell out the THC in abbreviation and introduction sections.**

**RESPONSE:** We have now spelt out the THC abbreviation in the introduction and abbreviation section.

## References

Agarwal S, Uchida Y, Mittapalli RK, Sane R, Terasaki T, and Elmquist WF. 2012. Quantitative proteomics of transporter expression in brain capillary endothelial cells isolated from P-glycoprotein (P-gp), breast cancer resistance protein (Bcrp), and P-gp/Bcrp knockout mice. *Drug Metabolism and Disposition* 40:1164-1169.

Doran A, Obach RS, Smith BJ, Hosea NA, Becker S, Callegari E, Chen C, Chen X, Choo E, and Cianfrogna J. 2005. The impact of P-glycoprotein on the disposition of drugs targeted for indications of the central nervous system: evaluation using the MDR1A/1B knockout mouse model. *Drug Metabolism and Disposition* 33:165-174.

Ejsing TB, Pedersen AD, and Linnet K. 2005. P-glycoprotein interaction with risperidone and 9-OH-risperidone studied in vitro, in knock-out mice and in drug-drug interaction experiments. *Human Psychopharmacology* 20:493-500.

Kirschbaum KM, Henken S, Hiemke C, and Schmitt U. 2008. Pharmacodynamic consequences of P-glycoprotein-dependent pharmacokinetics of risperidone and haloperidol in mice. *Behavioural Brain Research* 188:298-303.

Tang SC, de Vries N, Sparidans RW, Wagenaar E, Beijnen JH, and Schinkel AH. 2013. Impact of P-glycoprotein (ABCB1) and breast cancer resistance protein (ABCG2) gene dosage on plasma pharmacokinetics and brain accumulation of dasatinib, sorafenib, and sunitinib. *Journal of Pharmacology and Experimental Therapeutics* 346:486-494.

Vlaming M, Mohrmann K, Wagenaar E, de Waart DR, Elferink RO, Lagas JS, van Tellingen O, Vainchtein LD, Rosing H, and Beijnen JH. 2006. Carcinogen and anticancer drug transport by Mrp2 in vivo: studies using Mrp2 (Abcc2) knockout mice. *Journal of Pharmacology and Experimental Therapeutics* 318:319-327.

Wang JS, Ruan Y, Taylor RM, Donovan JL, Markowitz JS, and DeVane CL. 2004. The brain entry of risperidone and 9-hydroxyrisperidone is greatly limited by P-glycoprotein. *International Journal of Neuropsychopharmacology* 7:415-419.